# Active Thermal Control of IGBT Modules Based on Finite-Time Boundedness

**DOI:** 10.3390/mi14112075

**Published:** 2023-11-08

**Authors:** Zhen Hu, Xiaohua Wu, Man Cui

**Affiliations:** 1College of Automation, Nanjing University of Posts and Telecommunications, Nanjing 210023, China; 2School of Computer Science, Nanjing University of Posts and Telecommunications, Nanjing 210023, China; wuxiaohua@njupt.edu.cn; 3School of Information and Electronics, Beijing Institute of Technology, Beijing 100081, China; 7520210140@bit.edu.cn

**Keywords:** IGBTs, thermal management, reliability, junction temperature

## Abstract

One of the most important causes of the failure of power electronic modules is thermal stress. Proper thermal management plays an important role in more reliable and cost-effective energy conversion. In this paper, we present an advanced active thermal control (ATC) strategy to reduce a power device’s thermal stress amplitude during operation, with the aim of improving the reliability and lifetime of the conversion system. A state-space model based on a Foster-type thermal model is developed to achieve junction temperature estimation in real time. A feedback controller based on finite-time boundedness (FTB) is proposed to precisely regulate the temperature in order to reduce the thermal stress according to the temperature profile. The designed controller permits the precise control of the temperature and strongly reduces the thermal stress during fast transients in the power demand. Simulation and experimental results are provided to validate the effectiveness of the proposed method.

## 1. Introduction

As a core component of the power converter, insulated gate bipolar transistor (IGBT) modules have been widely employed in high-reliability and safety-critical systems, such as electric vehicles [1,2,3], aircraft [4], renewable energy generation [5,6], and high-speed railway [7]. However, owing to the challenging thermal environment combined with the aggressive power density, there may be huge temperature fluctuations in the IGBT modules, which may lead to severe thermal stress. Thermal stress may deteriorate the module’s electrical specifications and cause different degrees of thermo-mechanical failure, leading to reliability issues in power electronic applications. Research has shown that more than 30% of power conversion system breakdowns are caused by the power device failure. Moreover, nearly 60% of device failures are induced by thermal stress [8,9,10,11]. Therefore, thermal management has become a significant issue in power conversion systems from the point of view of reliability [12,13].

A large number of research articles on the thermal management of IGBTs have been presented in the past. For example, temperature monitoring provides an efficient approach to evaluating prototypes and further to limiting the device’s operational temperature to its threshold value; hence it is a feasible approach to intensifying the reliability of power conversion systems. Existing IGBT temperature monitoring approaches may be divided into optical methods, electrical methods, and physically contacting methods [14,15,16,17,18,19,20,21,22,23,24]. These studies focus on ensuring that the mean operational temperature (i.e., $T_{m}$) keeps below a safety threshold value during the full-charge conditions. Nevertheless, the reliability tests on the power devices indicate that a power device’s fatigue lifetime depends on thermal cycling, which is the temperature swing within devices caused by power cycles, i.e., loads. Thermal cycling strength can be characterized by the mean operational temperature ($T_{m}$) and the amplitude of temperature fluctuations (△T), as shown in Figure 1 [25,26,27].

Due to the mismatch of coefficients of thermal expansion of the various material layers typically used in devices and other packages, the bond wires and the solder layers subjected to thermomechanical stresses cause bond wire degradation and solder joint fatigue, i.e., thermal damage. Power devices fail when thermal damage surpasses the threshold. Moreover, as a matter of fact, △T has a greater impact on the device failure than $T_{m}$. From Figure 1, it can be concluded that, if △T is reduced by the same amount that $T_{m}$ is increased by, a much higher number of cycles to failure can be achieved. Unfortunately, △T is not taken into consideration in temperature monitoring approaches. Consequently, the device is only able to “safely” operate continuously at the maximum allowable temperature, while the potentially damaging stresses due to △T in the module cannot be avoided.

To address the aforementioned issues, active thermal control (ATC) techniques are developed to control against $T_{m}$ and △T simultaneously. In practical applications, by means of regulating the module’s cooling system or power losses, ATC techniques can reduce the amplitude of temperature fluctuations and the mean level of temperature. Furthermore, ATC techniques do not need to change the design of the power conversion system, meaning this type of technique is cost-effective [28,29,30]. There have been many efforts focusing on ATC techniques, which can be categorized into dynamic-cooling approaches and electrical-parameter approaches.

The dynamic-cooling approach to performing ATC depends on the active control of the cooling system. An advanced dynamic cooling strategy is proposed to reduce the module’s thermal cycling during the operation by controlling the speed of the fan or the flow velocity of the water-cooling system, aiming to improve the reliability performance of the power converter [31,32,33]. This type of technology can be employed in any system with controllable cooling.

The electrical-parameter approach to achieving ATC relies on the active control of the electrical parameters, which have a direct or indirect influence on the power losses generation or distribution of the module. Generally, the control strategy of the electrical-parameter approach may be categorized into three control levels: modulation level, converter level, and system level. In the modulation level, the gate driver and modified modulation patterns are always utilized to regulate the module’s power losses [34]. In the converter level, the thermal cycling of the modules can be reduced by modifying some controllable variables, such as DC-link, current, and switching frequency [35,36]. At the system level, the presence of multiple power converters can be utilized to adjust the losses distribution without disturbing the main converter’s goal [37,38].

Although both the dynamic-cooling approach and the electrical-parameter approach are competent at reducing the module’s thermal cycling to improve the reliability of the power converter effectively, there are still some limitations in practical applications: (a) an output observer for the information of $T_{m}$ and △T is essential to provide the feedback control signal, making the system costly and complex, and (b) the magnitude of $T_{m}$ and △T can only be reduced, while it cannot be controlled with precision based on the applications and the desired profiles. The above weaknesses may make these methods conservative. Consequently, designing a feedback control system independent of an observer for controlling $T_{m}$ and △T with precision remains challenging and is vital to improve the reliability of power electronic applications.

Motivated by the analysis described above, in this paper, we propose a novel feedback control system based on the theory of finite-time boundedness (FTB), which is able to precisely control the $T_{m}$ and △T of a power device without notably affecting the normal converter’s operation. Two aspects of the work are demonstrated: (a) a state-space thermal model of the power device is built, which is able to obtain the temperature information (i.e., △T and $T_{m}$) in real-time according to the electrical variables of the converter, and (b) a feedback controller based on FTB is proposed to precisely control $T_{m}$ and △T. Through the approach in this paper, the power losses of the module are regulated and the thermal stress is controlled; thereby, the damage caused by the thermal cycling is reduced, improving the reliability performance of the converter. Compared to the traditional ATC techniques, this approach has two advantages: (a) without using an observer, an accurate real-time estimate of junction temperature for a power device is still available and (b) the values of $T_{m}$ and △T can not only be reduced but also assigned precisely by the feedback controller.

The remainder of the paper is organized as follows. In Section 1, the development of a state-space thermal model of a power device is demonstrated. In Section 2, the feedback controller based on the FTB is introduced. In Section 3 and Section 4, the effectiveness of the proposed method is validated by simulation and experimental results, respectively.

## 2. Development of a State-Space Thermal Model

An accurate real-time estimation of junction temperature for the power device is an important part of the ATC algorithms. One way to obtain the temperature information is the use of integrated sensors. Negative temperature coefficient (NTC) resistors and on-chip diodes are the two common types of sensors. Generally, NTC resistors are installed in the direct-bond-copper (DBC) substrate to acquire the baseplate temperature [39], and on-chip diodes are integrated within the IGBT chip itself to perform online measurement of the chip temperature [40]. However, during the design and manufacturing of the power device, both types of sensors require some special considerations, such as electrical isolation, the layout and/or compatibility of pins, which may lead to the increase of manufacturing cost and induce some new reliability problems.

Another way to acquire the temperature information is the use of thermo-sensitive electrical parameters (TSEPs) in the power device, such as turn on/off time, on-state collector-emitter voltage, short-circuit current, and peak gate current. However, these electrical parameters face many difficulties in online implementation; for instance, the demand to compensate for the operation conditions and the demand for a high-precision measurement circuit or the redesign of the converter structure. Therefore, an economic and straightforward temperature estimation approach is significant for the ATC of a power device. In this paper, we proposed a state-space thermal model for the real-time estimation of the junction temperature according to the thermal behavior of the module.

### 2.1. Modeling

Typically, a power module is composed of IGBT chips and diode chips, which act as the heat sources that contribute to the entire heat flow inside the module. Figure 2 demonstrates a commercial power module (SKM75GB123D) made by SEMIKRON, where one IGBT chip and one free-wheeling diode chip are combined in parallel on each substrate tile. Generally, the temperature of an IGBT chip on one single substrate tile is influenced by the adjacent diode chips. In this paper, due to the long distance between the IGBT chip and the diode chip, cross-coupling is relatively small and, thereby, is ignored. As a result, only the self-heating of the IGBT chip is considered in the modeling process.

The IGBT chip, acting as the heat source, contributes to the entire heat inside the device. The heat is generated in the chip and spreads through several layers with different materials down to the baseplate, constituting the thermal path of the device, as illustrated in Figure 3. Most of the heat spreads down along an angle of 45°, regarded as the optimal thermal path. This thermal path can be described by the transient thermal impedance from the junction to ambient $Z_{JA}\left(t\right)$, which is shown as follows:(1)$$Z_{JA}\left(t\right)=\left(T_{J}\left(t\right)-T_{A}\right)/P,$$
where $Z_{JA}\left(t\right)$ denotes the transient thermal impedance from junction to ambient, *P* denotes the total power losses of the module, $T_{J}\left(t\right)$ denotes the junction temperature, and $T_{A}$ denotes the ambient temperature.

It is worth highlighting that the variations of ambient temperature are generally slower in comparison to the thermo-dynamic characteristics of the power device, and the ambient temperature normally remains constant via the dynamic cooling system (e.g., by controlling the speed of the fan). Therefore, the ATC of $T_{J}\left(t\right)$ is equivalent to the ATC of $\left(T_{J}\left(t\right)-T_{A}\right)$, and $\left(T_{J}\left(t\right)-T_{A}\right)$ could be described as $T_{J}^{*}\left(t\right)$ for model simplification. Thus, (1) can be rewritten as follows:(2)$$Z_{JA}\left(t\right)=T_{J}^{*}\left(t\right)/P.$$

The function of $Z_{JA}\left(t\right)$ can be described by an electrical equivalent resistance-capacitance (RC) network, shown in Figure 4, which is known as the Foster network. A series of exponential terms is used to characterize the time response of the Foster network as follows:(3)$$Z_{JA}\left(t\right)=\sum_{i=1}^{n}R_{i}\left(1-e^{-t/R_{i}C_{i}}\right),$$
where $R_{i}$ and $C_{i}$ denote thermal resistance and thermal capacitance of the electrical equivalent network.

Taking the Laplace transformation of (Equation 3), the partial fraction expansion form of the transfer function of $Z_{JA}\left(t\right)$ in the frequency domain is obtained as follows:(4)$$Z_{JA}\left(s\right)=\sum_{i=1}^{n}\frac{k_{i}}{s+p_{i}},$$
where $k_{i}$ and $p_{i}$ denote the residues and poles of the transfer function, respectively, and *s* denotes the complex variable.

According to the algebraic transformation, it was found that poles and residues have the relationships with the RC components as follows:(5)$$k_{i}=\frac{1}{C_{i}},\;p_{i}=\frac{1}{R_{i}C_{i}}.$$

It should be pointed out that there is no correlation between the RC elements and the physical characteristics of the thermal path, since the Foster network is just an equivalent circuit model of the module’s thermal system.

The partial fraction expansion shown in (Equation 4) can be easily transformed into a state-space model, as shown in the following form:(6)$$\left\{\begin{array}{cc} \dot{x}\left(t\right) & =\mathrm{Ax}(t)+Bu(t),\;(\mathrm{state}\;\mathrm{equation})\\ T_{J}^{*}\left(t\right) & =\mathrm{Cx}(t),\;\;\;\;(\mathrm{output}\;\mathrm{equation}), \end{array}\right.$$
where the state vector x(t) denotes the heat through the thermal path, u(t)=P(t) is the input of the thermal system, and P(t) is the power losses of the power module. The output equation gives the temperature of the system $T_{J}^{*}\left(t\right)$, which is the junction temperature of the power module.

Based on the relationships between residues, poles, and RC elements, the state-space model can be transformed into a parallel form with a diagonal system matrix, as shown in the following [41]:(7)$$\left\{\begin{array}{c} \begin{array}{cc} A & =\left(\begin{array}{ccccc} -\frac{1}{R_{1}C_{1}} & 0 & 0 & \cdots & 0\\ 0 & -\frac{1}{R_{2}C_{2}} & 0 & \cdots & 0\\ 0 & 0 & -\frac{1}{R_{3}C_{3}} & \cdots & 0\\ \vdots & \vdots & \vdots & \ddots & \vdots \\ 0 & 0 & 0 & \cdots & -\frac{1}{R_{n}C_{n}} \end{array}\right),\\ B & ={\left(\begin{array}{ccccc} 1 & 1 & 1 & \cdots & 1 \end{array}\right)}^{T},\\ C & =\left(\begin{array}{ccccc} \frac{1}{C_{1}} & \frac{1}{C_{2}} & \frac{1}{C_{3}} & \cdots & \frac{1}{C_{n}} \end{array}\right), \end{array} \end{array}\right.$$
where $A_{n\times n}$ is the system matrix, $B_{n\times 1}$ is the input matrix, and $C_{1\times n}$ is the output matrix.

### 2.2. Identification of Model Parameters

The state-space thermal model of a power device is given by (Equation 7), and the model’s parameters may be extracted for temperature estimation. As can be seen in (Equation 7), the matrices of the state-apace model are composed of RC parameters of the Foster network; thereby, the only thing we need to do is to acquire the RC parameters.

As an equivalent circuit model, the parameters of the Foster network are fitted from the transient thermal impedance $Z_{JA}\left(t\right)$. Meanwhile, $Z_{JA}\left(t\right)$ can be easily obtained by finite element analysis (FEA). A transient thermal analysis of the model system based on the dimensions and materials of the device and heat sink is processed by a commercial FEA software, i.e., ANSYS.

The design of the thermal analysis is implemented as follows. (a) The heat sink is cooled by forced-air convection, and the cooling surface keeps constant at 25 °C. (b) The IGBT module operates in a full-bridge inverter, as shown in Figure 5. The total power losses of the IGBT module are composed of conduction loss and switching loss, and can be estimated as follows [28,42,43,44]:(8)$$\left\{\begin{array}{cc} P & =P_{cond}+P_{sw},\\ P_{cond} & =V_{CE-ON}\times I_{C},\\ P_{sw} & =\left(E_{on}+E_{off}\right)\times f_{sw}, \end{array}\right.$$
where *P* denotes the total power losses of the module, $P_{cond}$ denotes the conduction power loss, $P_{sw}$ denotes the switching power loss, $I_{C}$ is the collector current, $V_{CE-ON}$ is the on-state collector-emitter voltage, $E_{on}$ and $E_{off}$ represent the turn-on and turn-off energy of the module, respectively, and $f_{sw}$ is the switching frequency. Based on the operation conditions in Table 1, the power losses of the module are estimated by (Equation 8). (c) The thermal analysis is processed in ANSYS under a transient mode for 10 s and the sampling interval is 0.001 s. Placing the power losses on the IGBT chips, the results of the thermal analysis for the IGBT module are obtained, as shown in Figure 6. At the same time, we set up the thermal analysis experiment platform. In the experiment, the IGBT module proceeded with the same operating conditions as the simulation, and the temperature distribution map of the upper surface of the device was obtained by an infrared camera, as shown in Figure 6. From Figure 6, we can see that the temperature distribution results of the simulation and experiment are consistent. The slight temperature difference may be caused by the difference in heat dissipation conditions.

By substituting the simulation results into (Equation 1), the transient thermal impedance curves $Z_{JA}\left(t\right)$ for the IGBT are derived, as shown in Figure 7. It has been found that a fourth-order Foster network has a good approximation for the transient thermal impedances. Adopting the least-square fitting method, the values of the RC component are obtained and shown in Table 1.

This way, the state-space thermal model has been built and can achieve a temperature estimate in real time according to the power loss, which depends on the electrical variables. In contrast to the traditional temperature measurement method, the proposed method can be commonly applied to any type of power device, and has economic and convenient advantages.

## 3. State Feedback Controller Based on FTB

The traditional ATC approach, which depends on simple feedback regulation, can reduce the thermal cycling, while it cannot control △T and $T_{m}$ with precision. In this paper, a state feedback controller based on the theory of FTB is proposed, which can precisely control the values of △T and $T_{m}$.

### 3.1. Theory of FTB

In this section, we deal with the concept of FTB. A system is considered an FTB if, once a time interval is fixed, its state does not exceed some bounds during this time interval [45,46,47,48]. That is to say, the thermal cycle of the power device can be fixed in a given boundary during operation based on the theory of FTB.

Considering the state-space model (Equation 6), it represents the thermal system of the power device and can be turned in to a normal form (i.e., a time-invariant linear system) as follows:(9)$$\left\{\begin{array}{cc} \dot{x}\left(t\right)= & Ax\left(t\right)+Bu\left(t\right)+Gw\left(t\right),\;x\left(0\right)=x_{0}\\ y(t)= & Cx(t), \end{array}\right.$$
where $A\in R^{n\times n}$, $B\in R^{n\times m}$, $G\in R^{n\times r}$, and $C\in R^{p\times n}$. *w*(t) is the constant disturbance and $w^{T}\left(0\right)w\left(0\right)\leq \delta_{w}^{2}$.

Given system (Equation 9), we consider the state feedback controller:(10)u(t)=Kx(t),
where $K\in R^{m\times n}$.

In this section, we will introduce the sufficient conditions that guarantee that the system given by (Equation 9) and (Equation 10) (i.e., the thermal system of the power device) is bounded during the operation of the power converter. Generally, the concept of FTB can be described as the following definition.

**Definition 1.** 
*Finite-Time Boundedness.The time-invariant linear system (Equation 9) is said to be finite-time-bounded with respect to $\left(\delta_{x},\delta_{w},\epsilon ,R,T\right)$, where R is a positive-definite matrix, $0\leq \delta_{x}<\epsilon$, $\delta_{w}>0$, if*

(11)
$$\left.\begin{array}{cc} \; & x^{T}\left(0\right)Rx\left(0\right)\leq \delta_{x}^{2}\\ \; & w^{T}\left(0\right)w\left(0\right)\leq \delta_{w}^{2} \end{array}\right\}\Rightarrow x^{T}\left(t\right)Rx\left(t\right)<\epsilon^{2},\;\forall t\in \left[0,T\right].$$



Based on the definition of FTB, from (Equation 9) and (Equation 11), it can be seen that:(12)||x(t)||≤ϵ⇒||y(t)||≤||C||·ϵ;
for a specific system, the value of ||C|| is constant and ϵ can be selected according to the practical applications and the desired profiles. Thus, the output of the system is norm-bounded. That is to say, if system (Equation 9) satisfies the definition of FTB, the output of the system will never exceed the given boundary.

Next, we will show the sufficient conditions for system (Equation 9) with a state feedback controller (Equation 10) that is finite-time-bounded with respect to $\left(\delta_{x},\delta_{w},\epsilon ,R,T\right)$.

**Theorem 1.** 
*Finite-Time Boundedness via State Feedback. System (Equation 9) is finite-time-bounded with respect to $\left(\delta_{x},\delta_{w},\epsilon ,R,T\right)$, if there exist positive-definite matrices $Q_{1}\in R^{n\times n}$ and $Q_{2}\in R^{r\times r}$ and a positive scalar α such that the following conditions hold:*

(13)
$$\left(\begin{array}{cc} AQ_{1}+Q_{1}A^{T}+BL+L^{T}B^{T}-\alpha Q_{1} & GQ_{1}\\ Q_{1}G^{T} & -\alpha Q_{2} \end{array}\right)<0,$$


(14)
$$\frac{\delta_{x}^{2}}{\lambda_{min}\left(\tilde{Q_{1}}\right)}+\lambda_{max}\left(Q_{2}\right)\delta_{w}^{2}<\frac{\epsilon^{2}}{e^{\alpha t}\lambda_{max}\left({\tilde{Q}}_{1}\right)},$$

*where $\tilde{Q_{1}}=R^{1/2}Q_{1}R^{1/2}$. In this case, the controller K is given by $K=LQ_{1}^{-1}$.*


The proof of Theorem 1 has been added to Appendix A.

### 3.2. ATC of Thermal Cycle Based on FTB

The main aim of ATC is to precisely control $T_{m}$ and △T to reduce the thermal damage of the power device. In this note, a state feedback controller based on FTB is proposed to control the values of $T_{m}$ and △T simultaneously.

Firstly, we consider the original thermal system of the power device:(15)$$\left\{\begin{array}{cc} \dot{x}\left(t\right) & =Ax(t)+Bu(t),\\ T_{J}^{*}\left(t\right) & =Cx(t), \end{array}\right.$$
and the target thermal system of the power device:(16)$$\left\{\begin{array}{cc} \dot{\hat{x}}\left(t\right) & =0,\\ {\hat{T}}_{J}^{*}\left(t\right) & =C\hat{x}\left(t\right), \end{array}\right.$$
where $\hat{x}$ is the state of the target system and ${\hat{T}}_{J}^{*}\left(t\right)$ is the output of the target system. Define the error between the two systems $e\left(t\right)=x\left(t\right)-\hat{x}\left(t\right)$ and the controller $u\left(t\right)=K[x\left(t\right)-\hat{x}\left(t\right)]$. For the time period t∈[0,T], the error system can be written as
(17)$$\begin{array}{cc} \dot{e}\left(t\right) & =\dot{x}\left(t\right)-\dot{\hat{x}}\left(t\right)=\dot{x}\left(t\right)\\ \; & =Ax(t)+Bu(t). \end{array}$$

Recalling that $e\left(t\right)=x\left(t\right)-\hat{x}\left(t\right)$ and $u\left(t\right)=K[x\left(t\right)-\hat{x}\left(t\right)]$, we have the following closed-loop system:(18)$$\begin{array}{c} \dot{e}\left(t\right)=\left(A+BK\right)e\left(t\right)+A\hat{x}\left(t\right). \end{array}$$

Letting $\bar{A}=A+BK$, the error system can be rewritten as:(19)$$\begin{array}{c} \dot{e}\left(t\right)=\bar{A}e\left(t\right)+A\hat{x}\left(t\right), \end{array}$$
where $\hat{x}\left(t\right)$ can be regarded as the constant disturbance. Define $e^{T}\left(0\right)Re\left(0\right)\leq \delta_{e}^{2}$ and ${\hat{x}}^{T}\left(0\right)\hat{x}\left(0\right)\leq \delta_{\hat{x}}^{2}$. Based on the *Theorem of FTB*, a controller *K*, which guarantees that system (Equation 19) is finite-time-bounded with respect to $\left(\delta_{e},\delta_{\hat{x}},\epsilon ,R,T\right)$, is obtained. Thus, we have
(20)$$\begin{array}{c} e^{T}\left(t\right)Re\left(t\right)<\epsilon^{2} \end{array}\Rightarrow ||e\left(t\right)||<\epsilon .$$

Simple calculations show that $e\left(t\right)=x\left(t\right)-\hat{x}\left(t\right)$ implies
(21)$$\left.\begin{array}{cc} Ce(t) & =C[x\left(t\right)-\hat{x}\left(t\right)]\\ \; & =Cx\left(t\right)-C\hat{x}\left(t\right)\\ \; & =T_{J}^{*}\left(t\right)-{\hat{T}}_{J}^{*}\left(t\right) \end{array}\right\}\Rightarrow \left||C|\right|\cdot \left||e\left(t\right)|\right|\geq \left|\right|T_{J}^{*}\left(t\right)-{\hat{T}}_{J}^{*}\left(t\right)\left|\right|.$$

Putting together (Equation 20) and (Equation 21), we have
(22)$$\left|\right|T_{J}^{*}\left(t\right)-{\hat{T}}_{J}^{*}\left(t\right)||<||C||\cdot \epsilon ,$$
where ${\hat{T}}_{J}^{*}\left(t\right)$ represents ($T_{m}-T_{A}$), which is equal to ($T_{J}^{*}\left(t\right)-T_{A}$) in the ideal situation, while $T_{A}$ is constant, and ||C||·ϵ represents △T.

As can be seen from (Equation 22), the output of the thermal system (i.e., $T_{J}^{*}\left(t\right)$) swings at ${\hat{T}}_{J}^{*}\left(t\right)$ within the boundary of ||C||·ϵ. Therefore, the values of $T_{m}$ and △T are precisely controlled on the basis of state controller *K*. While considering the redundancies in the system as well as the potential costs for higher loss, ${\hat{T}}_{J}^{*}\left(t\right)$ and ϵ should be selected according to the practical applications and the desired profiles.

## 4. Simulation Validation

In this section, the effectiveness of the proposed method, which is able to precisely control the values of $T_{m}$ and △T, is validated by a numerical analysis. The control scheme using the electrical variables to adjust the power losses to control the module’s thermal cycling is shown in Figure 8.

In the control scheme, the switching frequency $f_{sw}$, load current $I_{C}$, and DC-link voltage $V_{DC}$ are collected by the physical system. In addition, the device’s electrical characteristics are taken into account to identify the voltage $V_{CE-ON}$ and the switching energies $E_{on}$ and $E_{off}$. These electrical parameters are used to calculate module’s power losses, *P*, which are presented to the state-space model to estimate the module’s temperature. The feedback controller u(t)=Kx(t) is utilized to precisely regulate the electrical variables according to the set value of $T_{m}$ and △T. In this section, the electrical variables, including $I_{C}$ and $V_{DC}$, are set to be constant to simplify the control complexity, while the switching frequency $f_{sw}$ is selected as the only variable to regulate the power loss for temperature control.

The design of the simulation analysis is performed as follows to eliminate the influence of various operation conditions: (a) The heat sink is water-cooling, and the water-cooling runners are inside the heat sink, as shown in Figure 9. The heat sink material is aluminum, and the water-cooling runners keep the heat sink temperature constant at 25 °C; (b) the DC-link voltage $V_{DC}$ is 100 V, and the load current $I_{C}$ is sinusoidal current, as shown in Figure 10; (c) the basic value of the switching frequency $f_{sw}$ is 10 kHz, while it can vary from 5 kHz to 20 kHz; (d) the converter modulation frequency $f_{0}$ is 10 Hz; (e) the simulation analysis is processed in ANSYS under a transient mode for 10 s. The results of the simulation test are described next.

Firstly, the effectiveness of the state-space model, which is to achieve a temperature estimate according to the power losses of the device, is demonstrated. The power module used in this section is shown in Figure 2, and the power losses are calculated with (Equation 8). The state-space model, which is composed of (Equation 6) and (Equation 7) with the parameters in Table 1, estimates the temperature on the basis of currents, shown in Figure 10. Meanwhile, the FEA of the power module is processed in ANSYS to acquire the temperature according to the same load currents. The junction temperature $T_{J}$ estimated by the state-space model is compared with the results from FEA, which is shown in Figure 11.

The $T_{J}$ estimate via the state-space model is consistent with the FEA results during the various operation conditions. The correlation coefficient $\gamma^{2}$ between the two results is more than 0.95, and the maximum error is about 1.2 °C or 1.6% of the total range for each waveform. The difference may be linked to errors inherent in the modeling process. This indicates that the state-space model can accurately estimate the junction temperature in real time.

Additionally, the converter modulation frequency also affects the junction temperature response. Hence, the modulation frequency of the converter is varied to demonstrate the the consistency of the $T_{J}$ estimate under various operation conditions. The $T_{J}$ estimate during the modulation frequency of 5 Hz and 10 Hz is shown in Figure 12. The $T_{J}$ estimate via the state-space model agrees with the FEA results. It can be seen that △T in $T_{J}$ is reduced with the increase of frequency, whereas the mean temperature $T_{m}$ remains constant. This is attributed to the frequency response of the thermal system replicating a low-pass filter.

The results described above indicate that, without using an observer, the temperature is still accurately obtained via the state-space model during various operating conditions.

Secondly, the effectiveness of the feedback controller based on FTB, which can precisely control the temperature, is illustrated. The results described in Figure 11 clearly show that the thermal cycle △T is large enough, which will lead to severe thermal damage and accelerate the fatigue of the module. Thus, the thermal cycle needs to be reduced for the improvement of the power device’s reliability.

Recall from Section 3.2, the values of △T and $T_{m}$ should be properly selected on the basis of the practical applications and desired profiles to obtain the controller *K*. In this section, the values of △T and $T_{m}$ are selected according to the load currents in Figure 10 and the temperature results in Figure 11, where the modulation frequency is 10 Hz. The values of △T and $T_{m}$ are set to be 3 °C and 60 °C, respectively. Considering (Equation 19), we have
(23)$$\left\{\begin{array}{cc} \left|\right|T_{J}^{*}\left(t\right)-{\hat{T}}_{J}^{*}\left(t\right)\left|\right| & =||△T||\leq ||C||\cdot \epsilon ,\\ \left|\right|{\hat{T}}_{J}^{*}\left(t\right)\left|\right| & =\left|\right|T_{m}-T_{A}\left|\right|\\ \; & \leq \left||C|\right|\cdot \left|\right|\hat{x}\left(t\right)\left|\right|\\ \; & \leq \left||C|\right|\cdot \delta_{\hat{x}}. \end{array}\right.$$

According to (Equation 23), the values of ϵ and $\delta_{\hat{x}}$ are acquired and used to calculate the controller *K*. Based on the *Theorem of FTB*, the controller *K* is obtained by solving the linear matrix inequality (LMI) of (Equation 13) and (Equation 14) and is $K_{1}=\left[-24.2197,-14.4039,-101.7307,-153.8223\right]$. The controller is able to change the electrical variable (i.e., switching frequency in this section) to regulate the power losses for ATC.

The designed controller test during various operation conditions has been carried out by simulation. Based on the mission profiles shown in Figure 10, the temperature results with and without control are demonstrated in Figure 13. Compared to temperature results without control, the designed controller can reduce the amplitude of temperature fluctuations and the mean temperature effectively. The variations of $T_{m}$ and △T are shown in Table 2.

According to the results in Table 2, the designed controller strongly reduced the temperature fluctuations during a fast changing power demand, and it almost eliminates the effect of the varying power loss profiles. Moreover, the mean temperature with control is equal to the set value (i.e., 60 °C), and the temperature variations never exceed the boundary of 3 °C. This indicates that the designed controller is able to precisely control the temperature.

The simulation results with the realistic load profiles demonstrate the ability of the designed controller to precisely control △T and $T_{m}$. As a consequence, the thermal stress of the devices can be reduced and the lifetime can be extended by the active thermal controller. Moreover, the controller can be adapted for any multi-layer structured power device to extend the operational reliability.

## 5. Experimental Validation

In this section, the effectiveness of the proposed method is further exhibited by an experimental study. The experimental scheme, which consists of a power converter formed by the IGBT module shown in Figure 2 (the packaging of one IGBT module is intentionally removed), a control system to process the ATC of the IGBT module, a gate driver to generate gate signals for the IGBT module, a DC power supply for the test currents, an IR camera to measure the junction temperature of IGBT, and an aluminum heat sink to cool the IGBT module, is illustrated in Figure 14.

Adopting the control strategy in the simulation section, the switching frequency $f_{sw}$ is selected as the only variable to regulate the module’s power losses in order to precisely control the junction temperature, and the test conditions are set as follows: (a) the heat sink is cooled by forced-air convection, and the temperature of the bottom surface keeps constant at 25 °C; (b) the DC-link voltage $V_{DC}$ is constant, and the test current $I_{C}$ is sinusoidal current, shown in Figure 8; (c) the basic value of switching frequency $f_{sw}$ is 10 kHz, while it can change from 5 kHz to 20 kHz; (d) the experimental analysis is processed in a transient mode for 300 s. The results of the experimental test are described next.

Firstly, the effectiveness of the state-space model, which is proposed to estimate the junction temperature, is demonstrated. As described in Section 2.2, the parameters of *R* and *C* have been obtained using the FEA method and are shown in Table 1. The electrical variables, including $V_{DC}$, $f_{sw}$, and $I_{C}$, are collected and utilized to calculate the power loss of the module according to (Equation 8). The state-space model, which is composed of (Equation 6) and (Equation 7) with the parameters in Table 1, estimates the junction temperature based on the calculated power losses. Meanwhile, the IR camera is used to measure the junction temperature. The temperature results from the state-space model and the IR camera are shown in Figure 15.

It is obvious that the $T_{J}$ estimate from the state-space model tracks the $T_{J}$ measurement from the IR camera accurately during the various operation conditions with a maximum error of 2.8%. The maximum difference between the two signals is located at the peaks of the temperature profile, and is about 2.2 °C, which may be linked to the errors inherited from the modeling process and/or measurement noise from the IR camera. The results indicate that the state-space model can accurately obtain the junction temperature information during various operation conditions.

Secondly, the effectiveness of the feedback controller based on FTB, which is able to precisely control the temperature, is illustrated. The temperature results shown in Figure 15 show that the thermal cycle △T has reached 10 °C, which will lead to severe thermal stress and, thereby, needs to be controlled to reduce the thermal damage.

As described in Section 3.2, the values of △T and $T_{m}$ should be properly selected based on the practical applications and desired profiles to calculate the controller *K*. In this section, the values of △T and $T_{m}$ are set according to the temperature results in Figure 15, and are consistent with the setting of simulation section. The values of △T and $T_{m}$ are 3 °C and 60 °C, respectively, and are used to calculate ϵ and $\delta_{\hat{x}}$ by (Equation 23). Substituting ϵ and $\delta_{\hat{x}}$ in (Equation 13) and (Equation 14), the thermal controller *K* is obtained, which is $K_{1}=\left[-24.2197,-14.4039,-101.7307,-153.8223\right]$.

The test of the designed controller during various operation conditions has been performed experimentally. The test load condition profiles are based on the currents shown in Figure 10. The test temperature results with and without control are shown in Figure 16. Compared to the temperature results without control, the designed controller is competent at reducing the module’s thermal cycling due to the variations of current profiles. The values of △T and $T_{m}$ are illustrated in Table 3.

As can be seen, the value of △T reduces greatly from 9.4 °C to 2.36 °C, and the value of $T_{m}$ reduces from 67.3 °C to 59 °C. In addition, the value of $T_{m}$ with control is approximate to the set value, and the value of △T never exceeds the given boundary of 3 °C. It should be noted that Figure 16 only shows the test currents running at a fixed value of $V_{DC}$, but different $V_{DC}$ and $I_{C}$ combinations have also been tested and receive very similar, good results.

To further demonstrate the effectiveness of the proposed method, the realistic reliability improvements made by decreasing thermal damage are presented. Typically, the reliability performance of the IGBT module can be indicated by the evolution of on-state collector-emitter voltage $V_{CE-ON}$. The evolution of $V_{CE-ON}$ can be realized using a power cycling test. The power cycling tests during the operation conditions with/without ATC are processed respectively, and the values of $V_{CE-ON}$ under these two test conditions are measured continuously. Then, the test results are obtained and are shown in Figure 17. Compared with the test results with ATC, the values of $V_{CE-ON}$ without ATC have a larger growth during the power cycling test, and the difference between the two voltage signals increases monotonously, meaning that the IGBT module without ATC has greater thermal damage. This phenomenon indicates that, in practical applications, the reliability performance of IGBT can be improved by the proposed ATC method.

## 6. Discussion

The experimental results with a realistic load profile clearly show that the proposed controller is effective in precisely regulating the values of △T and $T_{m}$ to reduce the thermal stress. The on-state collector-emitter voltage $V_{CE-ON}$ is a common parameter used to evaluate the device fatigue. Compared with the results with ATC, the values of $V_{CE-ON}$ without ATC have a greater growth under the same operation conditions. Moreover, the difference between the two voltage signals increases monotonously, indicating that the thermal damage rate without ATC accelerated in service. Consequently, the reliability performance of the power conversion system improved with the proposed control strategy. In this paper, the switching frequency $f_{sw}$ is selected as an electrical variable to adjust the device’s power loss for the ATC implementation, while the variables, such as DC-link voltage $V_{DC}$, collector current $I_{C}$, and the modulation patterns, also have a significant influence on the device’s power loss. In practical applications, we can select appropriate variables based on working conditions to complete ATC control.

## 7. Conclusions

This paper proposed a novel ATC strategy based on FTB to reduce the power device’s thermal stress to achieve a conversion system with higher reliability and a longer lifetime. The ATC technique includes two parts: a state-space model and a feedback controller based on FTB. The state-space model estimates the junction temperature in real-time, and the feedback controller based on FTB regulates the values of △T and $T_{m}$ to reduce thermal damage. The simulation and experimental results validated the proposed control strategy. The fluctuations of temperature during fast-changing power demand are precisely controlled. As a result, the lifetime of the power converter was extended. This study will aid in the development of temperature control methods and the improvement of the reliability of the power converter in the future.

## Figures and Tables

**Figure 1 micromachines-14-02075-f001:**
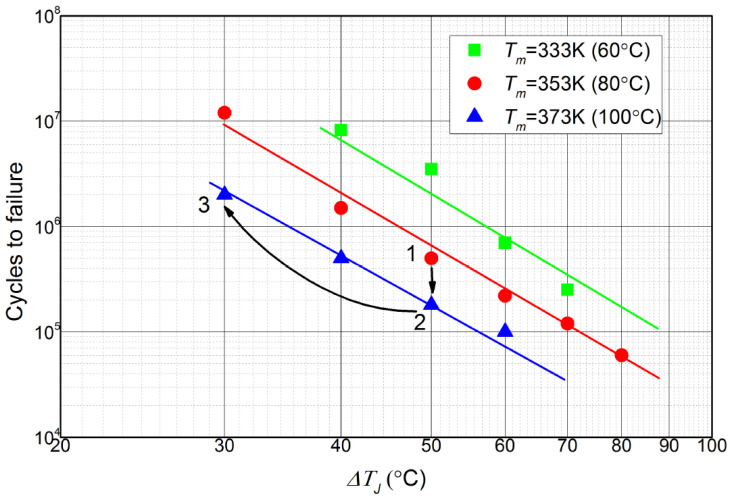
Results of reliability tests on the power devices.

**Figure 2 micromachines-14-02075-f002:**
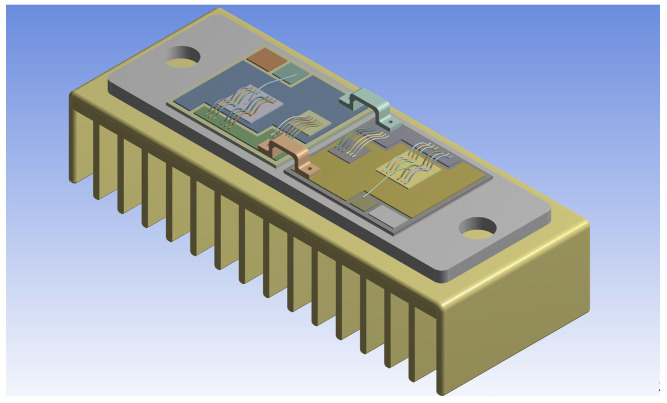
A commercial IGBT module.

**Figure 3 micromachines-14-02075-f003:**
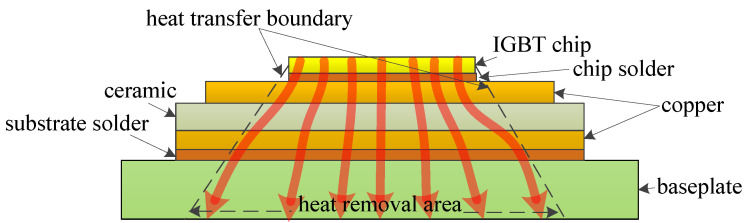
The thermal path of the IGBT module.

**Figure 4 micromachines-14-02075-f004:**
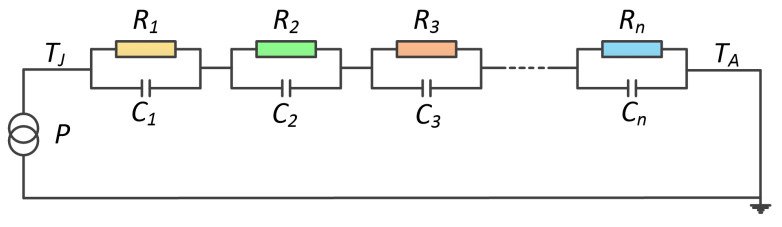
Foster-type network.

**Figure 5 micromachines-14-02075-f005:**
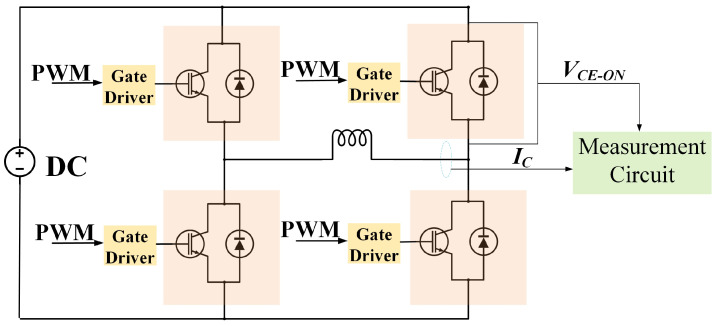
Block diagram of a full-bridge inverter.

**Figure 6 micromachines-14-02075-f006:**
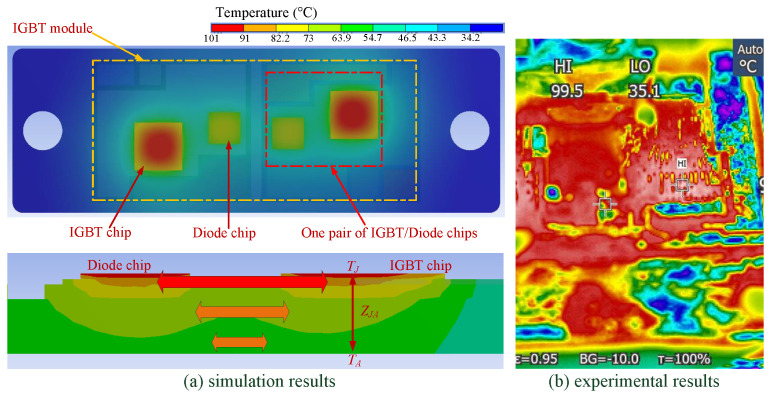
The results of thermal analysis for the IGBT module.

**Figure 7 micromachines-14-02075-f007:**
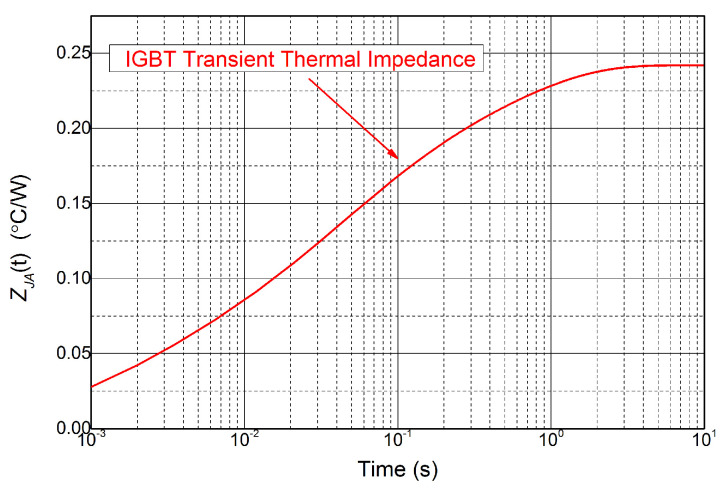
The transient thermal impedance curve of IGBT module.

**Figure 8 micromachines-14-02075-f008:**
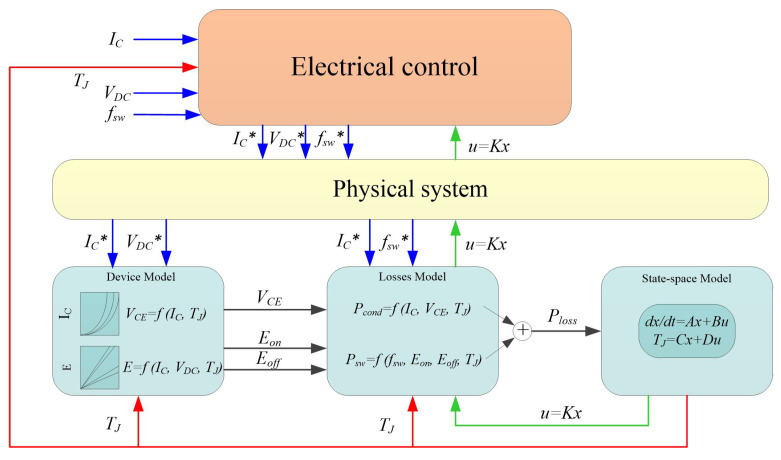
A model-based active temperature control scheme.

**Figure 9 micromachines-14-02075-f009:**
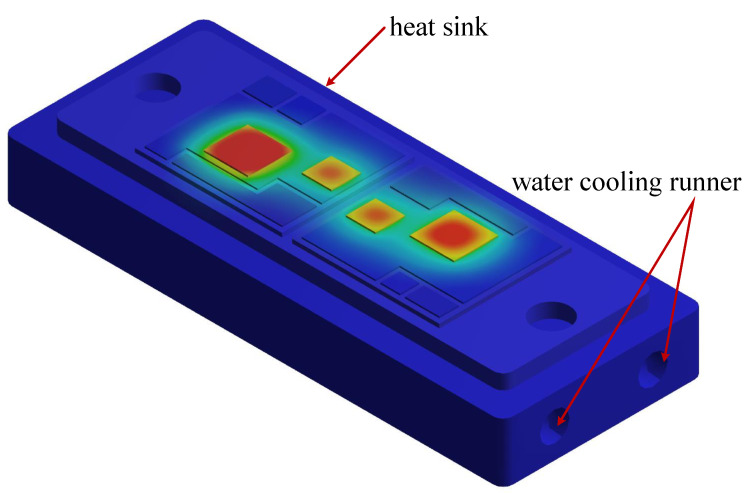
The water-cooling runners inside the heat sink.

**Figure 10 micromachines-14-02075-f010:**
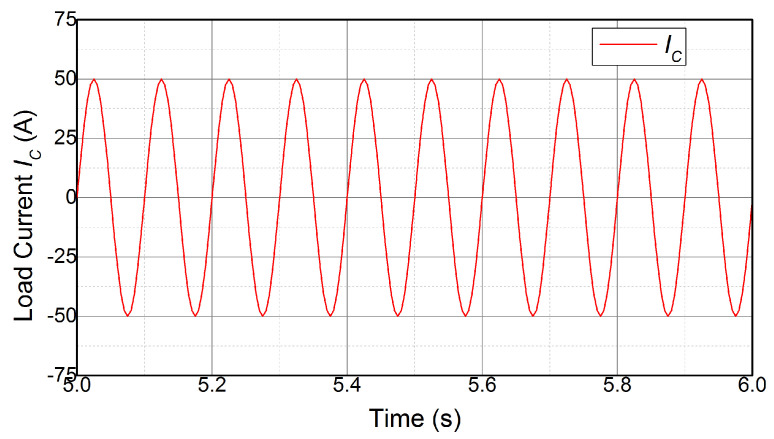
Load current.

**Figure 11 micromachines-14-02075-f011:**
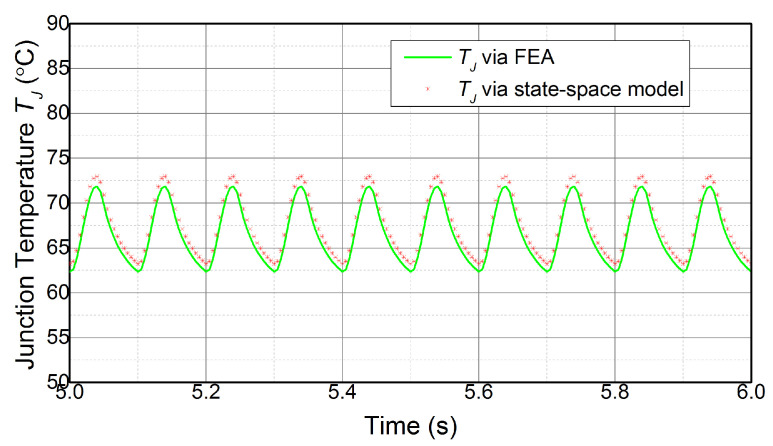
$T_{J}$ estimate by the state-space model compared to FEA results.

**Figure 12 micromachines-14-02075-f012:**
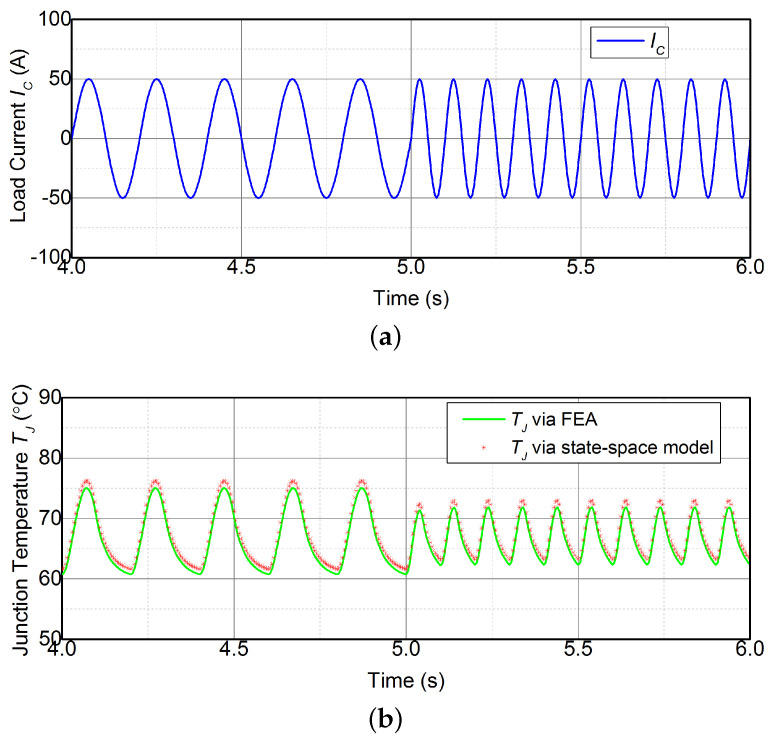
(**a**) Load currents and (**b**) FEA results and $T_{J}$ estimate during the modulation frequency of 5 Hz and 10 Hz.

**Figure 13 micromachines-14-02075-f013:**
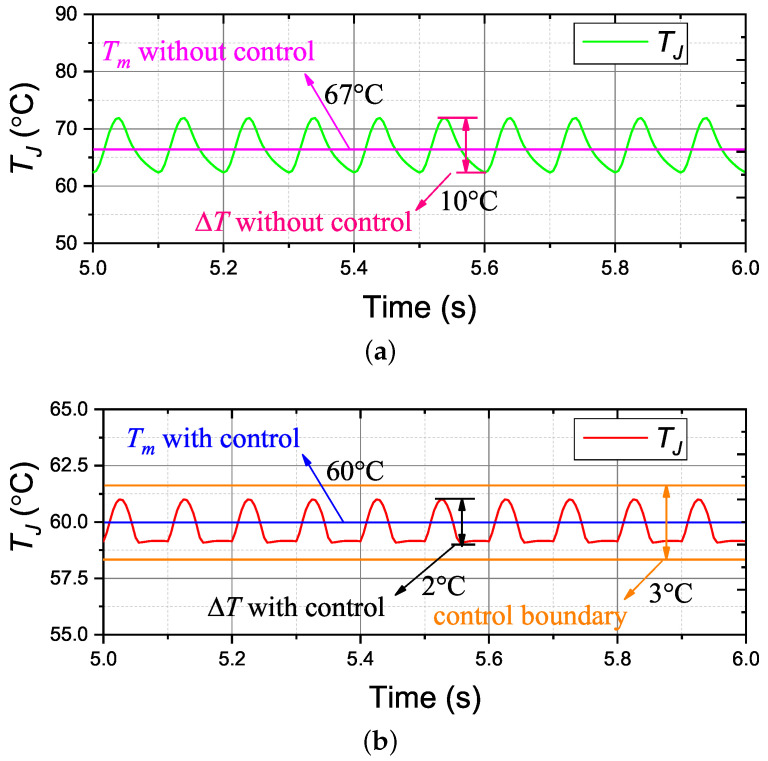
Simulation results of ATC with and without the controller $K_{1}$. (**a**) without the controller; (**b**) with the controller.

**Figure 14 micromachines-14-02075-f014:**
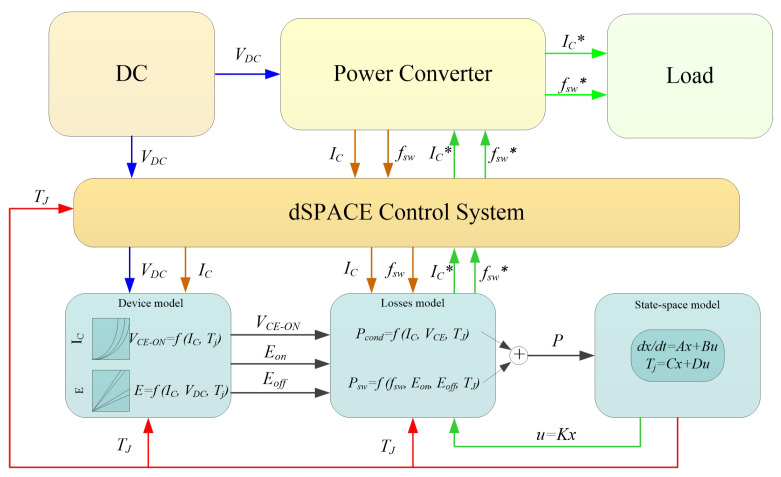
The scheme of experimental setup.

**Figure 15 micromachines-14-02075-f015:**
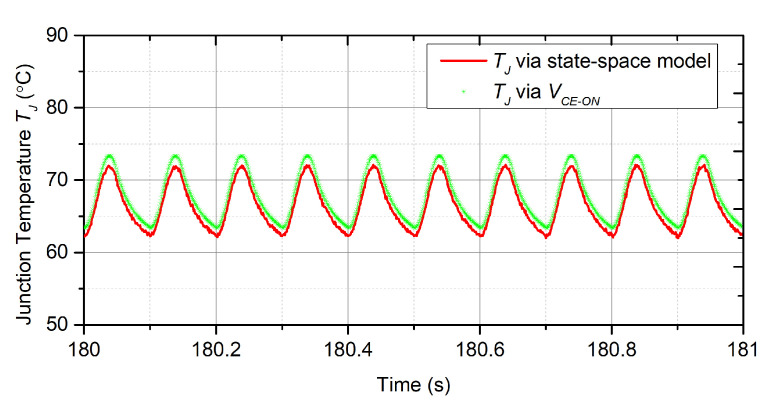
$T_{J}$ estimate by the state-space model compared to $V_{CE-ON}$ measurements.

**Figure 16 micromachines-14-02075-f016:**
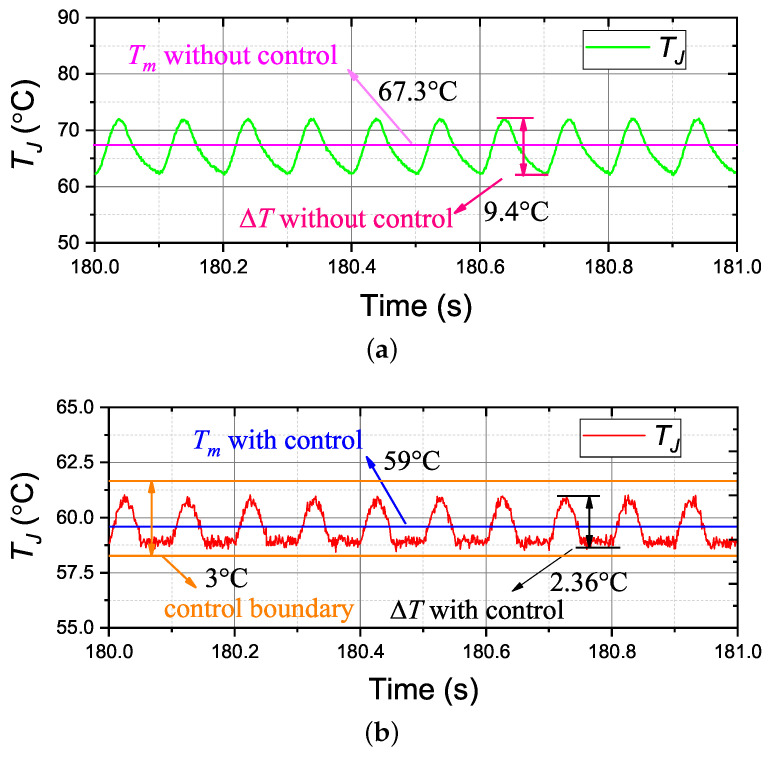
Experimental results of ATC with and without the controller $K_{1}$. (**a**) without the controller; (**b**) with the controller.

**Figure 17 micromachines-14-02075-f017:**
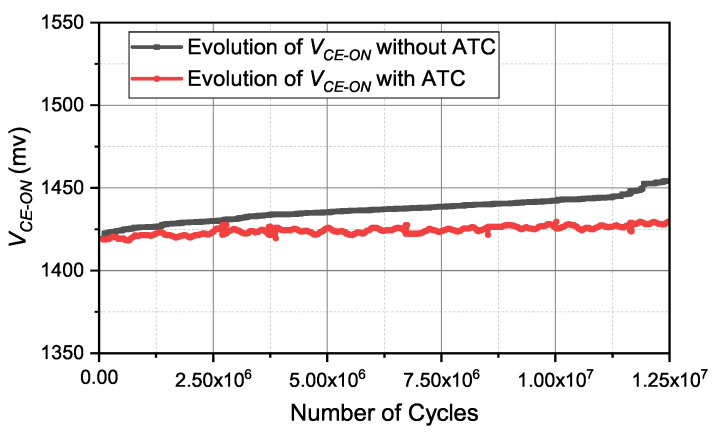
The evolution of $V_{CE-ON}$ during the power cycling test.

**Table 1 micromachines-14-02075-t001:** The identified model parameters.

*i*	1	2	3	4
$R_{i}$ (K/W)	0.18	0.064	0.022	0.004
$C_{i}$ (J/K)	0.182	0.75	0.36	1.25

**Table 2 micromachines-14-02075-t002:** Comparisons of temperature with and without control.

	With Control	Without Control
$T_{m}$ (°C)	60	67
△T (°C)	2	10

**Table 3 micromachines-14-02075-t003:** Comparisons of temperature with and without control.

	With Control	Without Control
$T_{m}$ (°C)	59	67.5
△T (°C)	2.36	9.4

## Data Availability

Not applicable.

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
