# Peer review of "Active Thermal Control of IGBT Modules Based on Finite-Time Boundedness"

_micromachines, 2023, doi:10.3390/mi14112075_

Round 1

Reviewer 1 Report

Comments and Suggestions for Authors

Reviewer’s comments on micromachines-2597812

In this manuscript, an active thermal control (ATC) strategy to reduce the thermal stress amplitude of power device during operation was proposed, which improved the reliability and lifetime of conversion system. Based on the results shown in the manuscript, I suggest making significant revisions to this work,

1. Please explain the definition of “thermal damage” (lines 225, 296, 359, 379, 388, 400) and “the damage caused by the thermal cycling” (line 85) in detail, respectively, and their hazards should be mentioned in the Introduction section.

2. The thermal path in Figure 3 was not clearly presented. The authors should add more detailed description and mark the path with obvious (maybe red) arrows in the figure.

3. The results in Figure 6 should be verified by those from infrared thermography, to show the comparison between the simulation results and the experimental ones at actual situation, and thermocouples should be applied to represent temperature changes at key temperature monitoring locations in the actual module.

4. Line 264, as for the sentence “(a) the heat sink is cooled by water-cooling, and the cooling surface keeps constant at 25℃”, the distribution of the water-cooling runners and schematic diagram of the heat sink should be provided.

5. Lines 406~408, as for the statement “while the variables such as DC-link voltage VDC, collector current IC, modulation patterns that have a significant influence on the device’s thermal cycling have not been present in this work.”, why did the authors put it in Conclusion and Discussion section? By the way, I suggest to revised the subtitle as “Conclusions”, and there are several grammatical mistakes in the manuscript, such as the “present” in the above sentence.

Comments on the Quality of English Language

Moderate editing of English language required. 

Reviewer 2 Report

Comments and Suggestions for Authors

In this paper, the authors present an advanced active thermal control (ATC) strategy to reduce a power device’s thermal stress amplitude during operation, with the aim of improving the reliability and lifetime of the conversion system. A Foster-type thermal model is developed.

This manuscript should deserve publication in MICROMACHINES after the addressing the following points.

- For many formulas, the parameters are not defined. For instance, what is P in (1)?

- physical meaning of parameters (with experimental values) should be exposed.

- line 13: it should be “1. Introduction” instead of “0. Introduction”

- line 31: refs [21-24] appear before [14-16] why? Please write “… past [14-24]” on line 27.

- “Discussion” section should be separated from “Conclusion”. Consequently, provide a discussion of the results and compare results with data from the literature.

- rewrite conclusion as section summarizing the present data.

Comments on the Quality of English Language

 Minor editing of English language required

Round 2

Reviewer 1 Report

Comments and Suggestions for Authors

Most of the comments have been addressed. 

Comments on the Quality of English Language

Moderate editing of English language required.